# Identification of a Cancer-Predisposing Germline *POT1* p.Ile49Metfs*7 Variant by Targeted Sequencing of a Splenic Marginal Zone Lymphoma

**DOI:** 10.3390/genes13040591

**Published:** 2022-03-26

**Authors:** Audrey N. Jajosky, Anna L. Mitchell, Mahmut Akgul, Shashirekha Shetty, Jennifer M. Yoest, Stanton L. Gerson, Navid Sadri, Kwadwo A. Oduro

**Affiliations:** 1Department of Pathology and Laboratory Medicine, University Hospitals Cleveland Medical Center, 11100 Euclid Avenue, Cleveland, OH 44106, USA; akgulm@amc.edu (M.A.); shashirekha.shetty@uhhospitals.org (S.S.); jennifer.yoest@uhhospitals.org (J.M.Y.); navid.sadri@uhhospitals.org (N.S.); 2Department of Pathology and Laboratory Medicine, University of Rochester Medical Center, 211 Bailey Road, West Henrietta, NY 14586, USA; 3Department of Genetics and Genome Sciences, University Hospitals Cleveland Medical Center, 11100 Euclid Avenue, Cleveland, OH 44106, USA; anna.mitchell@uhhospitals.org; 4M.D. Albany Medical Center, Department of Pathology, 43 New Scotland Avenue, Albany, NY 12208, USA; 5Department of Medicine, Hematology Oncology Division, University Hospitals Cleveland Medical Center, 11100 Euclid Avenue, Cleveland, OH 44106, USA; slg5@case.edu

**Keywords:** hereditary cancer predisposition, germline *POT1* alteration, splenic marginal zone lymphoma, cancer genetics

## Abstract

Germline disruptive variants in *Protection of Telomeres 1* (*POT1*) predispose to a wide variety of cancers, including melanoma, chronic lymphocytic leukemia (CLL), Hodgkin lymphoma, myeloproliferative neoplasms, and glioma. We report the first case of splenic marginal zone lymphoma (SMZL) arising in a patient with a germline *POT1* variant: a 65-year-old male with an extensive history of cancer, including melanoma and papillary thyroid carcinoma, who presented with circulating atypical lymphocytosis. Bone marrow biopsy revealed 20% involvement by a CD5^−^CD10^−^ B-cell lymphoma that was difficult to classify. During the clinical workup of his low-grade lymphoma, targeted next-generation sequencing (NGS) identified *POT1* p.I49Mfs*7 (NM_015450:c. 147delT) at a variant allele frequency (VAF) of 51%. NGS of skin fibroblasts confirmed the *POT1* variant was germline. This likely pathogenic *POT1* loss-of-function variant has only been reported once before as a germline variant in a patient with glioma and likely represents one of the most deleterious germline *POT1* variants ever linked to familial cancer. The spectrum of cancers associated with germline pathogenic *POT1* variants (i.e., autosomal dominant *POT1* tumor predisposition syndrome) should potentially be expanded to include SMZL, a disease often associated with the loss of chromosome 7q: the location of the *POT1* genetic locus (7q31.33).

## 1. Introduction

Utilization of targeted NGS in the clinical evaluation of lymphoid disorders is gaining traction. Genes that are somatically mutated in lymphoid malignancies, and rightfully included in clinical NGS panels, may also be mutated in familial cancer predisposition syndromes. *POT1* alterations occur in < 10% of sporadic CLL [1], but also in familial malignancies, including CLL [2], melanoma [3], Hodgkin lymphoma [4], colorectal carcinoma [5], and angiosarcoma [6]. We report the incidental discovery of a germline *POT1* variant during the diagnostic evaluation of a low-grade B-cell lymphoma that was difficult to classify by conventional morphologic and immunophenotypic studies.

## 2. Materials and Methods

### 2.1. Lymphoid NGS of Bone Marrow

Genomic DNA was extracted from the fresh diagnostic bone marrow aspirate using the MagNA Pure Compact Nucleic Acid Isolation Kit I and MagNA Pure Compact Instrument (Roche, Basel, Switzerland). The NGS library was prepared from 20 ng genomic DNA using the Ion Ampliseq Library Kit 2.0 (Thermo Fisher Scientific, Waltham, MA, USA). Template amplification and enrichment were performed using the Ion Chef and HiQ View sequencing kits (Thermo Fisher Scientific). NGS was performed on the Ion Torrent PGM (9 samples per v318 chip) sequencer (Thermo Fisher Scientific) using a custom in-house panel designed to detect single nucleotide variants and small insertions and deletions within 31 genes recurrently mutated in low-grade lymphoproliferative disorders: *ATM, BIRC3, BRAF, BTK, CARD11, CCND1, CD79B, CXCR4, EGR2, FBXW7, IKBKB, KLF2, KRAS, MAP2K1, MAP3K14, MYD88, NFKBIE, NOTCH1, NOTCH2*, *NRAS, PLCG2, POT1, RPS15, SF3B1, STAT3, STAT5B, TNFAIP3, TP53, TRAF2, TRAF3,* and *XPO1*. Reads were aligned to hg19. Variant calling was performed using the Torrent Variant Caller (v5.10) under somatic settings and a custom hotspot bed file containing all COSMIC (v87) entries spanning the targeted amplicons. All pathogenic and likely pathogenic variants with VAFs ≥ 2% were reported. A board-certified molecular pathologist interpreted variants according to the American College of Medical Genetics and Genomics (ACMG), Association for Molecular Pathology (AMP), American Society of Clinical Oncology (ASCO), and College of American Pathologists (CAP) guidelines.

### 2.2. NGS of Skin Fibroblasts for Confirmatory Germline Testing

Skin fibroblasts cultured from a left medial forearm biopsy were sent to Invitae for confirmatory germline testing using a targeted Multi-Cancer NGS panel (Test Code: 01101) that examines sequence changes and/or exonic deletions/duplications within 83 genes implicated in hereditary cancers: *ALK, APC, ATM, AXIN2, BAP1, BARD1, BLM, BMPR1A, BRCA1, BRCA2, BRIP1, CASR, CDC73, CDH1, CDK4*, *CDKN1B*, *CDKN1C, CDKN2A* (p14ARF)*, CDKN2A* (p16INK4a)*, CEBPA, CHEK2, CTNNA1, DICER1, DIS3L2, EGFR*, *EPCAM, FH, FLCN, GATA2, GPC3, GREM1, HOXB13, HRAS, KIT, MAX, MEN1, MET, MITF, MLH1, MSH2, MSH3, MSH6, MUTYH, NBN, NF1, NF2, NTHL1, PALB2, PDGFRA, PHOX2B, PMS2, POLD1, POLE, POT1, PRKAR1A, PTCH1, PTEN, RAD50, RAD51C, RAD51D, RB1, RECQL4, RET, RUNX1, SDHA, SDHAF2, SDHB, SDHC, SDHD, SMAD4, SMARCA4, SMARCB1, SMARCE1, STK11, SUFU, TERC, TERT, TMEM127, TP53, TSC1, TSC2, VHL, WRN,* and *WT1.* Genomic DNA was enriched for targeted regions using a hybridization-based protocol and sequenced using Ilumina technology.

### 2.3. Fluorescence In Situ Hybridization (FISH) Analysis of Bone Marrow Aspirate

Interphase FISH for trisomy 12 as well as deletions of 17p13.1 (*TP53*), 11q22.3 (*ATM*), 13q14.3, and 13q34 was performed on the bone marrow aspirate. A total of 250 nuclei were analyzed per probe.

## 3. Results

A 65-year-old man with a history of multiple primary tumors beginning in his 40s—including melanoma, prostatic adenocarcinoma, and papillary thyroid carcinoma—was found to have lymphocytosis during a routine medical visit. Peripheral blood showed abnormal small- to medium-sized lymphoid cells with moderately abundant cytoplasm, inconspicuous nucleoli and no overt hairy cytoplasmic projections (Figure 1A). A bone marrow biopsy revealed nodular non-paratrabecular lymphoid aggregates, of cells with a similar morphology, comprising 15–20% of total cells (Figure 1B). Based on immunophenotyping by flow cytometry and immunohistochemistry, the lymphoid cells were found to be lambda light chain-restricted B cells (CD20 bright) that expressed CD11c and partial CD103, but lacked CD5, CD10, CD23, CD200, CD123, CD25, Lef1, Cyclin D1, and Sox11. There was no identifiable plasmacytic differentiation. FISH identified a 17p (*TP53*) deletion in 20% of the bone marrow cells (Figure 1C). Lymphoid cells showed somatic mutation of the immunoglobulin heavy chain variable (*IGHV*) region (93.3% similarity to germline) and usage of the *IGHV3-53*04* allele. Clinical staging demonstrated new mild splenomegaly; no definite lymphadenopathy was reported. 

The main differential diagnosis for this hard-to-classify, low-grade B-cell lymphoma was SMZL versus hairy cell leukemia variant (HCL-v). Both can be challenging to diagnose because of a lack of disease-defining markers and reluctancy to biopsy the spleen, the primary site of disease, due to the associated morbidity. Molecular profiling, however, can aid in the diagnosis by identifying characteristic genetic differences. For example, unlike HCL-v [7,8], SMZL frequently harbors *NOTCH2* variants and mutations affecting the NF-κB signaling pathway [9].

To help classify the lymphoma, targeted Ion Torrent NGS was performed on the bone marrow aspirate using a custom 31-gene panel designed to evaluate low-grade lymphoproliferative disorders [10]. NGS identified two likely pathogenic variants: (1) *TRAF3* p.Lys168Glyfs*3 (NM_145725: c.501_502delGA) at a VAF of 8% and (2) *POT1* p.Ile49Metfs*7 (NM_015450: c.147delT) at a VAF of 51% (Table 1). Notably, *MAP2K1*, *BRAF*, *MYD88*, and *TP53* variants were not detected. TRAF3 is a negative regulator of the NF-κB signaling pathway, and the *TRAF3* VAF of 8% was consistent with the degree of bone marrow involvement by lymphoma. The overall clinical, morphologic, immunophenotypic, and molecular findings favored a diagnosis of SMZL.

In contrast to *TRAF3*, the *POT1* VAF of 51% was discordant with the frequency of lymphoma cells and suggestive of a potential heterozygous germline variant. Based on his strong history of cancer and discovery of the *POT1* variant, the patient was referred to a clinical geneticist by his oncologist for counseling and further evaluation. NGS of skin fibroblasts, performed using an orthogonal (Illumina) sequencing chemistry, confirmed the *POT1* p.Ile49Metfs*7 variant to be of germline origin. His family history revealed that many relatives—across multiple generations—developed lung, skin, and hematologic malignancies, including Hodgkin lymphoma and multiple myeloma (Figure 2A). Two of his relatives died from cancer at a young age: his father from mesothelioma at age 54 and his brother from lung cancer at age 44 (Figure 2A). Unfortunately, samples from relatives were not available for familial co-segregation studies. The overall findings, however, suggest that this germline early truncating *POT1* variant likely explains this individual’s remarkable personal and family history of cancer.

## 4. Discussion

*POT1* encodes one of six protein subunits of the shelterin complex that regulates telomere length and shields single-stranded telomeric DNA from being recognized as double-strand breaks. POT1 (Figure 2B) includes an *N*-terminal single-stranded DNA (ssDNA)-binding domain and a *C*-terminal protein-binding domain that anchors POT1 to the rest of the shelterin complex via telomere protection protein 1. Cancer-associated *POT1* alterations include diverse heterozygous loss-of-function variants believed to promote tumorigenesis in telomerase-expressing cells via telomere elongation, thereby extending the proliferative capacity and preventing senescence of cells [11]. In contrast, heterozygous *POT1* variants did not cause telomere deprotection, activation of a DNA-damage response, or genomic instability [11]. Therefore, oncogenic *POT1* variants predominantly appear to drive cancer through telomere lengthening that enhances the proliferative capacity of incipient tumor cells and enables acquisition of cooperating mutations required for cancer progression rather than through telomere deprotection and fragility [11]. In individuals with germline *POT1* variants, accumulation of distinct patterns of somatic alterations within tissue-specific progenitor cells likely underlies development of the vast array of cancers associated with the Li-Fraumeni-like *POT1* tumor predisposition syndrome. Identification of both telomeric and non-telomeric DNA-binding motifs within the *N*-terminal ssDNA-binding domain (specifically the OB-1 fold) of POT1 raises the possibility of a role beyond protection and maintenance of telomeres [12].

*POT1* p.Ile49Metfs*7 occurs within a region encoding the *N*-terminal OB-1 fold, creating a premature stop codon upstream of all functional domains (Figure 2B) that likely results in POT1 haploinsufficiency through protein truncation or nonsense-mediated messenger RNA decay. This alteration was previously identified as a germline variant in a patient with glioma [13]; however, this patient’s personal and family history of cancer are unknown, and no functional studies were performed to characterize the variant’s pathogenicity (communication with Dr. Victor Velculescu). The extensive personal and family history of cancer in this individual with six tumors, including SMZL, supports a likely oncogenic role for this *POT1* variant. Across the spectrum of germline *POT1* variants associated with cancer predisposition [14], p.Ile49Metfs*7 is the earliest frameshift mutation ever reported.

Recent studies implicate germline *POT1* alterations in predisposition to a wide variety of hematologic malignancies, including myeloproliferative neoplasms [15] and pediatric acute myeloid leukemia [16]. To our knowledge, this is the first reported case of SMZL arising in a patient with a germline *POT1* variant. Recurrent *POT1* alterations have not been described in multiple large-scale sequencing studies of SMZL, and we are not aware of any familial SMZL syndrome. Interestingly, *POT1* is located on 7q31.33, and hemizygous deletion of this chromosomal region, including *POT1*, represents the most common cytogenetic abnormality in SMZL, occurring in up to 40% of cases [17,18]. *TP53* (17p) deletion occurs in about 20–30% of SMZL cases [13]. Whether POT1 haploinsufficiency or telomere maintenance play a role in the pathogenesis of SMZL remains to be elucidated.

In summary, we: (1) utilized NGS to help classify a SMZL, (2) incidentally identified a germline *POT1* p.Ile49Metfs* variant, and (3) suggest this disruptive *POT1* variant underlies this individual’s extensive personal and family cancer history. As clinical tumor sequencing expands, germline variants in cancer susceptibility genes may be unexpectedly identified. Collaboration between pathologists, oncologists, and clinical geneticists is needed to optimally care for families affected by diverse forms of hereditary cancer predisposition.

## Figures and Tables

**Figure 1 genes-13-00591-f001:**
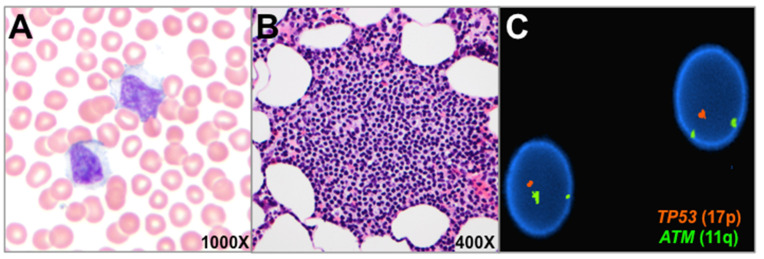
Blood and bone marrow involvement by SMZL with 17p deletion. Circulating atypical lymphocytes displayed condensed chromatin and moderately abundant pale blue cytoplasm (**A**). Lymphocytes lacked prominent cell surface projections and showed no evidence of plasmacytic differentiation. The bone marrow showed 20% involvement by a B-cell lymphoma that formed multiple ill-defined nodular aggregates (**B**). FISH demonstrated 17p (*TP53*) deletion in 20% of bone marrow cells (**C**).

**Figure 2 genes-13-00591-f002:**
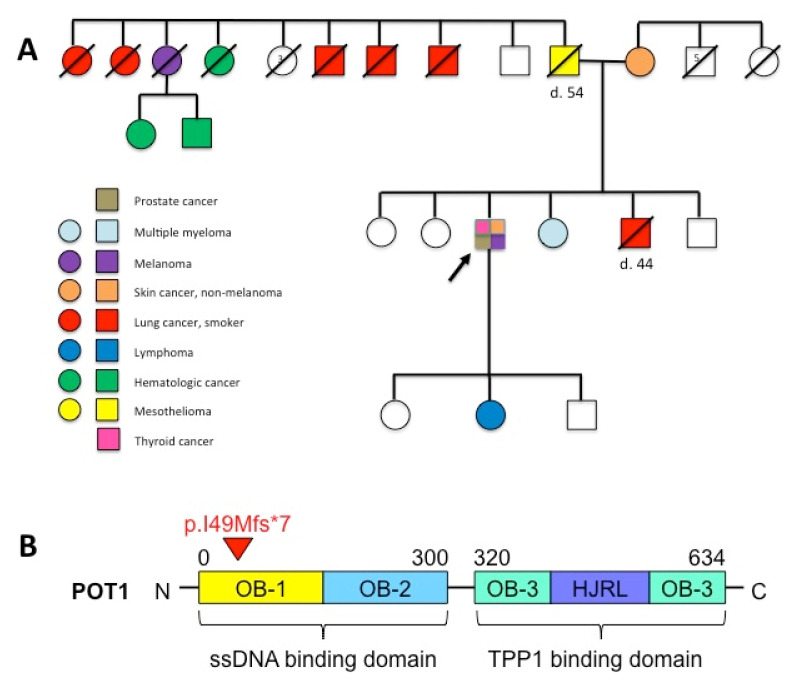
Family pedigree illustrating an extensive history of cancer and location of the *POT1* p.Ile49Metfs*7 variant. To date, the proband (arrow) has developed six primary tumors: melanoma (diagnosed at age 48), cutaneous squamous cell carcinoma (diagnosed at age 52), papillary thyroid carcinoma (diagnosed at age 55), cutaneous basal cell carcinoma (diagnosed at age 62), SMZL (diagnosed at age 65), and prostatic adenocarcinoma (diagnosed at age 66) (**A**). By patient report, almost all of the proband’s paternal relatives have developed cancer (**A**). A paternal aunt developed cervical cancer in her 50s and melanoma in her late 50s/early 60s; she also had two sons who developed leukemia in their 50s. An additional five paternal aunts and uncles, all of whom were smokers, developed lung cancer (ages at diagnosis unknown). Notably, the proband’s father died from mesothelioma at age 54 (age at diagnosis unknown), and his brother died from lung cancer at age 44 (diagnosed at age 42). The proband’s sister was diagnosed with multiple myeloma at age 54, and his daughter was diagnosed with Hodgkin lymphoma at age 27. (**A**). Only the proband’s son, who is unaffected by cancer, has been tested for the germline *POT1* p.Ile49Metfs*7 and found to be negative. (**B**). POT1 binds to the single-stranded G-rich telomeric overhang via its two *N*-terminal oligonucleotide/oligosaccharide binding (OB)-folds. The *C*-terminus contains another OB-fold and a Holliday junction resolvase-like (HJRL) domain which binds to telomere protection protein 1 (TTP1), anchoring POT1 to the shelterin complex. *POT1* p.Ile49Metfs*7 (red arrow) is the most upstream (5′) germline frameshift variant ever reported and leads to a premature stop codon in the seventh codon of the new reading frame (**B**).

**Table 1 genes-13-00591-t001:** NGS results and variant details.

Variant	VAF	Tissue Tested	Origin	COSMIC Database Frequency	ClinVar ID, gnomAD Population Frequency	Pathogenicity (ACMG Criteria)
*TRAF3* c.501_502delGA, p.Lys168Glyfs*3	8%	bone marrow	presumed somatic	absent	absent from ClinVar and gnomAD	likely pathogenic
*POT1* c.147delT, p.Ile49Metfs*7	51%	bone marrow and skin fibroblasts	confirmed germline	absent	ClinVar Variation ID: 420174; 0.007% (gnomAD)	likely pathogenic

## Data Availability

Invitae, who performed confirmatory germline NGS testing on skin fibroblasts, submitted this *POT1* p.Ile49Metfs*7 variant to ClinVar (Variant ID: 420174).

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
