# Peer review of "Identification of a Cancer-Predisposing Germline *POT1* p.Ile49Metfs*7 Variant by Targeted Sequencing of a Splenic Marginal Zone Lymphoma"

_genes, 2022, doi:10.3390/genes13040591_

Round 1
Reviewer 1 Report
The method is not described at all. The authors should provide information for the sequencing method/platform, read coverage, data analysis and the full gene list included in the bait set. Also the authors should present the results in an appropriate manner (e.g IGV snapshot, validation by Sanger sequencing).
Is the p.Ile49Metfs*7 variant classified as likely germline according to ACMG criteria or in silico prediction tools? Please clarify
The authors mention that fluorescent in situ hybridization 54 identified a 17p (TP53) deletion in 20% of marrow cells, but they do not present any FISH results.
Figure 1: Scale/magnification is missing.
The authors should provide a table with all different gene variants identified, allele frequency as well as if these alterations are reported in databases, e.g., cBioportal, Varsome, GnomeAD.
It would be interesting to provide more information for the history of the other family members ( age of diagnosis, metastasis if appeared)
Line 25-26: “NGS of skin fibroblasts confirmed the POT1 25 variant was germline”.
The family history indicates a high chance of a cancer predisposition gene, but analysis of skin fibroblast is not the best tissue for validation. The ideal germline sample would be blood or saliva.
Reviewer 2 Report
This study reports the detection of a genetic variant in POT1, p.I49Mfs*7 (NM_015450:c. 24147delT) in a male patient with an extensive history of different cancers. Through a family history study, the authors confirmed this genetic variant a potent germline pathogenic variants which can be linked to familial cancer. This is an interesting short communication with a significant clinical impacts. Some comments and suggestions may be helpful for the authors to improve the content of this report.
- It would be more solid if the authors can show the sequencing results in Fig.2, not just a summarized figure.
- Pls provide a proposed model and discuss the possible altered mechanisms associated this pathogenic POT1
- Pls check the writing format which should contain some sections like “Introduction” or “M & M”.
- Some additional somatic mutations may also contribute to the carcinogenesis of different cancer types in different family members, which should be clearly addressed in this report.
